# The role of the T-cell mediated immune response to Cytomegalovirus infection in intrauterine transmission

María Soriano-Ramos[1,2☯]*, Estrella Esquivel-De la Fuente[2☯], Eliseo Albert Vicent[3], María de la Calle[4], Fernando Baquero-Artigao[5], Sara Domínguez-Rodríguez[2], María Cabanes[6], Enery Gómez-Montes[7], Anna Goncé[8], Marta Valdés-Bango[8], Mª Carmen Viñuela-Benéitez[9], Mar Muñoz-Chápuli Gutiérrez[9], Jesús Saavedra-Lozano[10], Irene Cuadrado Pérez[11], Begoña Encinas[12], Laura Castells Vilella[13], María de la Serna Martínez[14], Alfredo Tagarro[15], Paula Rodríguez-Molino[5], Estela Giménez Quiles[3], Diana García Alcázar[7], Antonio García Burguillo[16], María Dolores Folgueira[17], David Navarro[3‡], Daniel Blázquez-Gamero[2,18‡], the CYTRIC Study Group[18¶]

1 Department of Neonatology, Hospital Universitario 12 de Octubre, Madrid, Spain, 2 Instituto de Investigación Hospital 12 de Octubre (imas12), Fundación Biomédica del Hospital Universitario 12 de Octubre (FBHU12O), Madrid, Spain, 3 Microbiology Department, Hospital Clínico Universitario de Valencia, Valencia, Spain, 4 Obstetrics Department, Hospital Universitario La Paz, Madrid, Spain, 5 Department of Infectious Diseases and Tropical Pediatrics, Hospital Universitario Infantil La Paz, Madrid, Spain, 6 Obstetrics Department, Hospital Universitario Príncipe de Asturias, Alcalá de Henares, Madrid, Spain, 7 Obstetrics Department, Fetal Medicine Unit, Hospital Universitario 12 de Octubre, Madrid, Spain, 8 Fetal Medicine Research Center, BCNatal - Barcelona Center for Maternal-Fetal and Neonatal Medicine (Hospital Clínic and Hospital Sant Joan de Déu), Institut Clínic de Ginecología, Obstetricia i Neonatologia, Institut d'Investigacions Biomèdiques August Pi i Sunyer, Universitat de Barcelona, Barcelona, Spain, 9 Obstetrics Department, Hospital Gregorio Marañón, Complutense University, Health Research Institute Gregorio Marañón, Madrid, Spain, 10 Hospital General Universitario Gregorio Marañón, Pediatric Infectious Diseases Unit, Universidad Complutense, Madrid, Spain, 11 Department of Neonatology, Hospital Universitario de Getafe, Madrid, Spain, 12 Obstetrics Department, Hospital Universitario Puerta de Hierro, Majadahonda, Madrid, Spain, 13 Department of Neonatology, Hospital Universitari General de Catalunya, Grupo Quiron Salud, Sant Cugat del Vallès, Barcelona, Spain, 14 Department of Neonatology, Hospital Infanta Sofía, San Sebastián de los Reyes, Madrid, Spain, 15 Paediatrics Department, Paediatrics Research Group, Hospital Universitario Infanta Sofía, Universidad Europea de Madrid, Madrid, Spain, 16 Obstetrics Department, Hospital Universitario 12 de Octubre, Madrid, Spain, 17 Microbiology Department, Hospital Universitario 12 de Octubre, Madrid, Spain, 18 Pediatric Infectious Diseases Unit, Hospital Universitario 12 de Octubre, Universidad Complutense, RITIP, Madrid, Spain

☯ These authors contributed equally to this work.
‡ DN and DBG are co-senior authors on this work.
¶ Membership of the CYTRIC Study Group is provided in the Acknowledgments section.
* sorianoramosmaria@gmail.com

**Data Availability Statement:** Data is not publicly available because is protected by European GDPR.

## Abstract

### Introduction

Prognostic markers for fetal transmission of Cytomegalovirus (CMV) infection during pregnancy are poorly understood. Maternal CMV-specific T-cell responses may help prevent fetal transmission and thus, we set out to assess whether this may be the case in pregnant women who develop a primary CMV infection.

### Methods

A multicenter prospective study was carried out at 8 hospitals in Spain, from January 2017 to April 2020. Blood samples were collected from pregnant women at the time the primary

However, can be formally shared under a formal application and research proposal after institutional acceptance. Please send your proposal to the secretary of the 12 de Octubre Hospital Ethics Committee: María Ugalde; e-mail: mugalde. imas12@h12o.es.

**Funding:** Founders (Spanish Ministry of Science and Innovation Instituto de Salud Carlos III and the European Regional Development Fund) did not play any role in study design, data collection and analysis, decision to publish or preparation of the manuscript. No commercial company funded the study or played any role in in study design, data collection and analysis, decision to publish or preparation of the manuscript. Grants: 1. Grant PI 16/00807, to DBG, from Spanish Ministry of Science and Innovation Instituto de Salud Carlos III and co-funded by the European Regional Development Fund. 2. Grant 19/01333, to DBG, from Spanish Ministry of Science and Innovation Instituto de Salud Carlos III and co-funded by the European Regional Development Fund. 3. Grant INT20/00086 from Spanish Ministry of Science and Innovation Instituto de Salud Carlos III and co-funded by the European Regional Development Fund. 4. There is no additional external funding received for this study. 5. DBG received received fees from MSD as speaker in educational activities not related to the present study. 6. MSD is not a founder of the study. Sponsors websites: www. isciii.es https://ec.europa.eu/regional_policy/en/funding/erdf.

**Competing interests:** I have read the journal's policy and the authors of this manuscript have the following competing interests: DBG received fees from MSD as speaker in educational activities. None of the remaining authors have any conflict of interests to declare. This does not alter our adherence to PLOS ONE policies on sharing data and materials.

CMV infection was diagnosed to assess the T-cell response. Quantitative analysis of interferon producing specific CMV-CD8$^+$/CD4$^+$ cells was performed by intracellular cytokine flow cytometry.

## Results

In this study, 135 pregnant women with a suspected CMV infection were evaluated, 60 of whom had a primary CMV infection and samples available. Of these, 24 mothers transmitted the infection to the fetus and 36 did not. No association was found between the presence of specific CD4 or CD8 responses against CMV at the time maternal infection was diagnosed and the risk of fetal transmission. There was no transmission among women with an undetectable CMV viral load in blood at diagnosis.

## Conclusions

In this cohort of pregnant women with a primary CMV infection, no association was found between the presence of a CMV T-cell response at the time of maternal infection and the risk of intrauterine transmission. A detectable CMV viral load in the maternal blood at diagnosis of the primary maternal infection may represent a relevant biomarker associated with fetal transmission.

## Introduction

In high-income countries, around 50% of women of childbearing age are seronegative for Cytomegalovirus (CMV) [1]. However, 1–7% of these women will be infected by CMV every year, resulting in a prevalence of congenital infection of 0.14–0.7% [1, 2]. Despite the impact of CMV infection, and although it is considered the most common cause of congenital neurodevelopmental delay, several issues remain unclear. Transmission is thought to be dependent on multiple factors, such as maternal and fetal immune systems, placental factors, maternal viral, load and viral strain and the time of maternal infection [3, 4]. Timing of fetal infection is a key predictive factor for long term outcomes in children with congenital CMV, and severe sequelae are associated with fetal infection in the embryonic or early fetal period, mainly first trimester of pregnancy [5]. Risk of fetal infection during pregnancy is higher after a primary infection (32–40%) than a non-primary infection (1.4%) [4, 6] and pre-existing immune response does appear to provide some protection from fetal transmission. Nevertheless, it is still not possible to accurately predict if maternal infection will be transmitted to the fetus, and biomarkers currently available including IgG avidity index, have limited prognostic value.

Studies on transplant recipients have also documented the importance of the CMV-specific T-cell response for the control of viral infection [7]. CMV-specific memory T-cells stimulated with peptide pools of CMV proteins IE-1, IE-2, and pp65 in a cultured enzyme-linked immunospot (ELISPOT) assay after maternal primary infection were evaluated by Fornara et al. They found that a higher cultured ELISPOT response was associated with a lower risk of transmission to the fetus [7].

Lillery et al. investigated the specific lymphoproliferative response (LPR) and intracellular cytokine (interferon[IFN]–γ and interleukin [IL]–2) production during the first year after primary CMV infection in 49 pregnant women, finding that transmitter mothers presented a significantly delayed development of the CD4$^+$ T-cell LPR, compared with those who did not [8].

Also, they found that the level of CMV-specific memory T-cells during the first months after infection was significantly lower in mothers who were transmitters [9].

Similarly, Revello et al. analyzed specific CD4+ T-cells by cytokine flow cytometry and LPR among 74 pregnant women with primary CMV infection. This study showed that LPR to CMV was significantly lowered or delayed in transmitter mothers [10]. However, other studies have shown different results compared to the studies mentioned above. Eldar-Yedidia investigated IFN-γ secretion upon whole blood stimulation from 76 primary CMV-infected pregnant women, with either CMV-peptides or phytohemagglutinin (PHA)-mitogen. The main finding was that low IFN-γ relative response (<1.8%) strongly correlated with absence of transmission [3]. Saldan et al. studied CMV ELISPOT assays in 57 pregnant women with a primary infection, finding that an increase in CMV ELISPOT levels was associated with a higher risk of fetal transmission [11].

Hence, this study aimed to untangle the role of the maternal T-cell response upon diagnosis of a primary maternal infection on the risk of fetal CMV transmission.

## Methods

### Study design

A multicenter prospective study was carried out at 8 hospitals in Spain, from January 2017 to April 2020. Blood samples were collected from pregnant women when they were diagnosed with a primary CMV infection, and CMV-specific CD8+ T lymphocytes (directed against the CMV proteins pp65 and IE-1) that produce IFN-γ (CMV-CD8+IFN-γ) were quantified by intracellular cytokine flow cytometry.

All pregnant women with a primary CMV infection were included in the study and although CMV screening during pregnancy is currently not mandatory in Spain [12], some centers perform it on a routine basis (e.g., Hospital La Paz), usually in the first trimester (or at the time of referral). Women with clinical symptoms consistent with CMV infection, with CMV seroconversion or a positive IgM during pregnancy, or with abnormal findings in fetal ultrasound (US), were referred to tertiary centers for further assessment and follow-up. Women with any kind of immunodeficiency or those receiving immunosuppressive therapy were excluded from this study. The study data were collected and managed using the REDCap (Research Electronic Data Capture) application hosted at Instituto de Investigación Hospital Universitario 12 de Octubre (Imas12: Madrid, Spain), a secure, web-based application designed to support data capture for research studies [13].

### Sample collection and T cell quantification

Blood samples (5 mL) were obtained in sodium heparin tubes from all the pregnant women recruited onto this study as close to the time of infection as possible. Each sample was sent to Hospital Clínico Universitario de Valencia (Valencia, Spain), where they were processed within 24 hours. T-cells were quantified using an intracellular cytokine flow cytometry procedure commercialized by Beckton Dickinson (BD Fastimmune, BDBiosciences, San Jose, CA). Heparinized whole blood (0.5 ml) was simultaneously stimulated for 6 h with two sets of 15-mer overlapping peptides (11-mer overlap) encompassing Cytomegalovirus IE-1 and pp-65 proteins (JPT peptide Technologies GmbH (Berlin, Germany)) at a concentration of 1 μg/ml per peptide, in the presence of 1 μg/ml of costimulatory monoclonal antibodies (mAbs) to CD28 and CD49d. Appropriate positive (phytohemagglutinin) and negative controls were used. Samples mock-stimulated with phosphate-buffered saline (PBS)/dimethyl sulfoxide and costimulatory antibodies were run in parallel. Brefeldin A (10 μg/ml) was added for the last 4 h of incubation. Blood was then lysed (BD FACS lysing solution) and frozen at −80˚C until

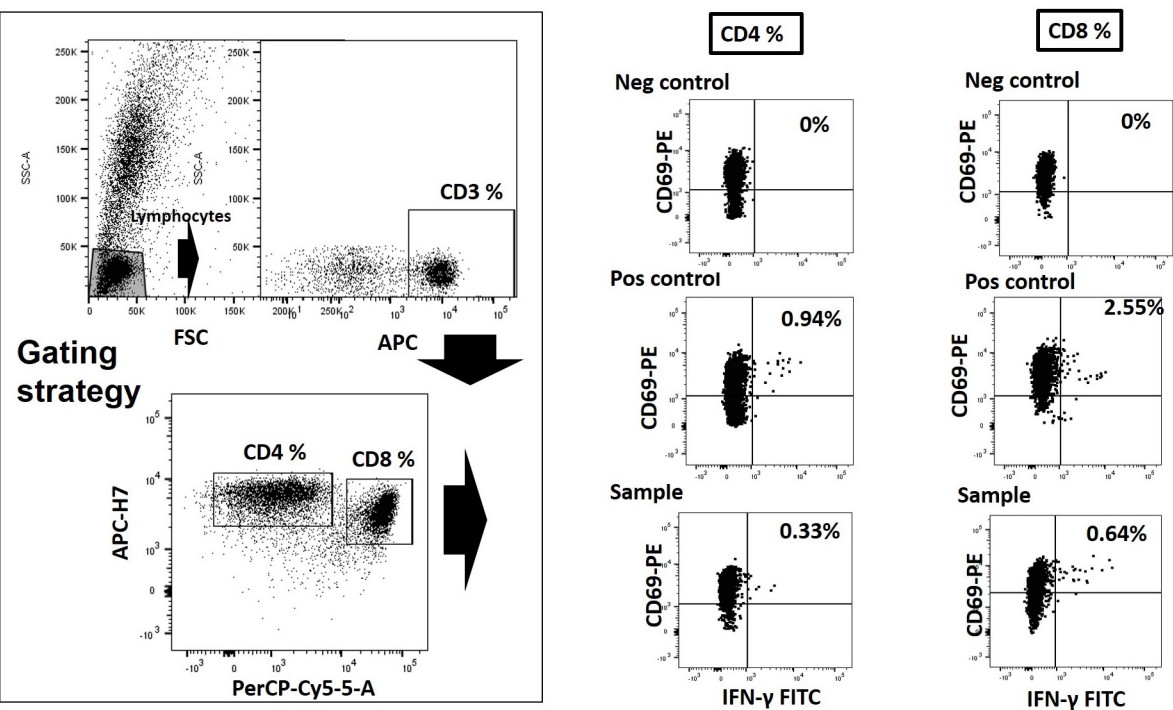

**Fig 1. Gating strategy for the enumeration of CMV-specific IFN-γ producing CD4+ and CD8+ T cells by Intracellular Cytokine Staining (ICS).** Total Lymphocyte/CD3+/CD8+ or Lymphocyte/CD3+/CD4+ events were gated and then analyzed for CD69+/IFN-γ production. Positive (Phytohemaglutinin) and negative controls (DMSO) were used.

tested. On the day of testing, stimulated blood was thawed at 37˚C, washed, permeabilized (BD permeabilizing solution) and stained with a combination of labeled mAbs (anti-IFNγ-FITC, anti-CD4-APC-H7, anti-CD8-PerCP-Cy5.5, and anti-CD3-APC) for 1 h at room temperature. Cells were then washed, resuspended in 200 μL of 1% paraformaldehyde in PBS, and analyzed within 2 h on an FACSCanto flow cytometer (BD Biosciences Immunocytometry Systems, San Jose, CA) using FlowJo software. CD3+/CD8+ or CD3+/CD4+ events were gated and then analyzed for CD69+/IFN-γ production. All data were corrected for background CD69+/IFN-γ production and expressed as the percentage of cells producing CD69+/IFN-γ by the total CD8$^+$ or CD4$^+$ T cells (Fig 1).

Only those samples with more than 1,000 events through the CD69/IFNγ window were considered valid. The total number of CMV-specific CD8$^+$ and CD4$^+$ T lymphocytes producing IFN-γ was calculated based on the positive events, and the absolute CD8$^+$ and CD4$^+$ T lymphocyte count. Specific CD8 and CD4 responses against CMV were defined as >0.1%.

## Definitions

**Primary CMV infection.** Was defined as the presence of CMV seroconversion during pregnancy, or positive IgM and IgG detection with a low avidity index (<50%) [14].

**Intrauterine fetal infection was.** Diagnosed by detection of viral DNA by real-time Polymerase Chain Reaction (PCR), either in amniotic fluid (AF) or the newborn's urine (first 14 days of life). If the woman had not yet delivered, she was considered a non-transmitter if the real time-PCR in AF was negative.

**Fetal infection by CMV.** Was diagnosed by positive CMV-PCR detection in AF or fetal blood obtained by cordocentesis.

**Congenital CMV infection.** (cCMV) was considered if the infant presented a positive CMV-PCR in urine within the first 14 days of life.

**The limit of blood VL detection.** Differed among participating hospitals: 12 de Octubre (120 IU/mL), La Paz (1000 IU/mL), Clínic Barcelona (20 IU/mL), Gregorio Marañón (100 IU/mL), Puerta de Hierro (35 IU/mL), Getafe (120 IU/mL), Infanta Sofía (500 IU/mL), General de Catalunya (150 IU/mL). To assess the IgG avidity index, a chemiluminescence based Abbott Alinity diagnostic platform was used.

**Symptomatic fetal infection.** [15] was defined as the presence of abnormalities compatible with CMV infection detected by US and/or fetal magnetic resonance image (MRI), and/or abnormalities in fetal blood obtained by cordocentesis (thrombocytopenia and/or anemia).

**Symptomatic infection at birth.** Was defined as the presence of abnormal physical features (jaundice, petechiae/purpura, splenomegaly and/or hepatomegaly, hypotonia, seizures, paresis, or weak sucking), chorioretinitis, sensorineural hearing loss (SNHL), small for gestational age (SGA), thrombocytopenia (platelet count $<100\times10^3/\mu$L), elevated alanine aminotransferase levels ($>80$ IU/L), hyperbilirubinemia (direct bilirubin $> 2$ mg/dL), microcephaly or neuroimaging abnormalities compatible with cCMV in cranial US/MRI. Newborns who did not fulfil any of the aforementioned criteria after a complete evaluation at birth were considered asymptomatic cCMV. SGA was defined as a birth weight below a standard deviation (SD) of –2 for gestational age (GA) [16]. Microcephaly was defined as a head circumference below –2 SD for GA [16]. SNHL was defined as a hearing threshold $>25$dB when tested by Auditory Brainstem Response in either ear, and it was evaluated at birth and 12 months of age.

## Statistical analysis

The descriptive statistics of the patient's characteristics were presented as frequencies (n) and percentages (%) in the case of categorical variables, and the median and interquartile range (IQR) for continuous variables. A $\chi2$ test or Fisher's exact test (if the expected number in any cell was $<5$) was applied to assess the differences between groups of categorical variables, and a Mann-Whitney U test for continuous variables. A logistic regression analysis was used to assess the association between a positive response of CD8 and CD4 lymphocytes in IFN-γ production on intrauterine CMV transmission. The continuous variable, the percentage of CD8 and CD4 lymphocyte IFN-γ production, was converted into categorical data based on a cut-off point of 0.1%, as used elsewhere [17], considering values above 0.1% as the presence of specific CD8 and CD4 responses against CMV. The time of maternal infection was estimated from the relative avidity in a linear regression analysis calculated based on the technical data provided by the supplier of the avidity assay (Abbott, Architect CMV IgG Avidity). The multivariate models were adjusted based on any preventive treatment with hyperimmune globulin (HIG) and valacyclovir, and the time from maternal infection to CD4 and CD8 sampling. Backward stepwise elimination was applied to reach the final multivariate model and Akaike information criteria (AIC) were used to identify the best-fitting model. As a result, the odds ratio (OR) and associated 95% confidence interval (CI) were obtained for each adjusted univariate and multivariate model. A p-value $<0.05$ was considered statistically significant and R software (R Core Team, 2015) was used for analysis.

## Ethics

The study was approved by the Institutional Review Board at the Hospital 12 de Octubre (IRB number: 17/007). Written informed consent was requested from all women included for clinical data collecting (about their pregnancies and their newborns) and blood sampling.

All the procedures were in accordance with the Helsinki Declaration (1964, recently amended in 2008) of the World Medical Association.

## Results

Blood samples were collected from 135 pregnant women with suspected CMV infection (Fig 2), of which 23 (17%) were excluded because the blood sample (n = 9) or the total lymphocyte count was not available(n = 2), the sample could not be analyzed because the conditions were inappropriate(n = 11) or the fetal transmission status was unknown at the time of the analysis (n = 1). Of the remaining 112 women, 60 experienced a primary CMV infection (53.6%), 5 a non-primary infection (4.5%) and in 3 cases it was periconceptional (2.7%). Serological data were not available for the remaining 44 pregnant women (38.9%) and hence, they could not be classified as primary or non-primary infections.

Data from the group of 60 pregnant women with a primary CMV infection was analyzed (Table 1). Twenty-four of them were transmitters, while 36 were non-transmitters. The median GA at maternal infection was 10 weeks (IQR 5–15.5) and it was higher in transmitter mothers (12.5[8.75–19.0] vs. 8[3.50–13.0], p = 0.013). There were four terminations of pregnancy (6.7%), most due to CMV-related fetal abnormalities evident by US (3/4) but one due to confirmed fetal infection without detecting US abnormalities at that time. Most pregnancies reached full term (91%, 51/56) and the median GA at delivery was 39.0 weeks (IQR = 38.0–40.0). The lowest GA was $32^{+4}$ weeks, which was a threatened preterm labour secondary to premature rupture of membranes at 31 weeks. Interestingly, the presence of other children in the family attending day-care was more frequent in the transmitter group (56.5% vs 20%, p = 0.01).

The CMV-VL in blood was available for 29 women (29/60, 48.3%) and CMV was detected in 6 of them (6/29, 20.7%), all transmitters (p = 0.004). None of the women with an undetectable VL in blood transmitted the infection to the fetus. In non-transmitter mothers the blood VL was determined earlier in pregnancy (16.6[14.1–26.4] vs. 28.4[24.6–29.4] weeks).

Amniocentesis was performed on 55 women (Fig 2), and the AF of 19 (34.6%) returned a positive CMV-PCR. Congenital infection was confirmed at birth in all but one of these cases (18/19, 94.7%). In this unconfirmed case, the VL in AF was very low (63.5 IU/mL) and this mother was treated with a single dose of HIG after the procedure at 23 weeks' gestation. Among the 36 mothers from whom the AF gave a negative CMV-PCR (36/55, 65.5%), most

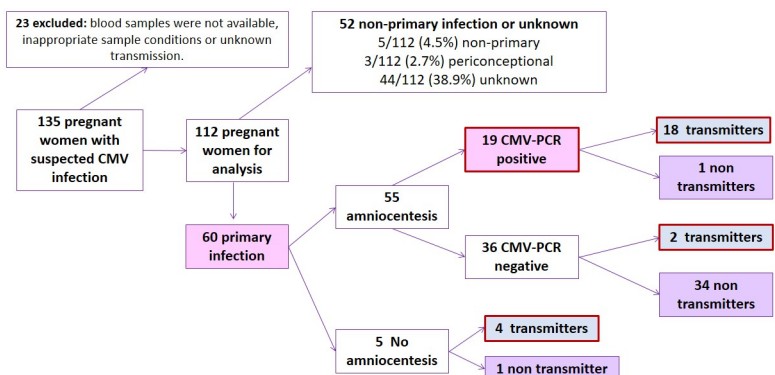

**Fig 2. Patient flowchart.** Flowchart showing outcome of 135 pregnant women with suspected Cytomegalovirus infection. CMV indicates Cytomegalovirus.

**Table 1.** Demographic data of the 60 women with a primary CMV infection during pregnancy studied here.

| | | | All women included (n = 60) | Non-transmitter (n = 36) | Transmitter (n = 24) | p-value |
|---|---|---|---|---|---|---|
| | | | n (%) | | | |
| **Age at diagnosis** | | | 34.0 [31.4,36.5] | 33.4 [30.2;36.5] | 34.8 [32.0,36.6] | 0.355 |
| **Reason for diagnosis** | | | | | | 0.292 |
| Maternal seroconversion | | | 19 (32.2) | 8 (22.9) | 11 (45.8) | |
| Positive IgM and IgG with low avidity index | | | 20 (33.9) | 14 (40.0) | 6 (25.0) | |
| Antenatal fetal US/MIR abnormalities | | | 2 (3.4) | 1 (2.9) | 1 (4.2) | |
| Maternal symptomatic infection | | | 10 (16.9) | 6 (17.1) | 4 (16.7) | |
| Contact with CMV infected person | | | 1 (1.7) | 1 (2.9) | 0 (0) | |
| Screening during pregnancy (serology) | | | 6 (10.2) | 5 (14.3) | 1 (4.2) | |
| **Children under 3 years** | | | 39 (66.1) | 23 (65.7) | 16 (66.7) | 1.000 |
| **Mother works with children** | | | 4 (7.1) | 2 (5.9) | 2 (9.1) | 1.000 |
| **Children in day-care** | | | 20 (34.5) | 7 (20.0) | 13 (56.5) | **0.010** |
| **Symptomatic infection** | | | 29 (48.3) | 18 (50.0) | 11 (45.8) | 0.958 |
| Lymphadenitis | | | 1 (1.7) | 1 (2.8) | 0 (0) | 1.000 |
| Infectious hepatitis | | | 6 (10.0) | 3 (8.3) | 3 (12.5) | 0.675 |
| Mononucleosis | | | 7 (11.7) | 5 (13.9) | 2 (8.3) | 0.691 |
| Fever | | | 14 (23.3) | 10 (27.8) | 4 (16.7) | 0.493 |
| Other | | | 11 (18.3) | 5 (13.9) | 6 (25.0) | 0.321 |
| **Time of primary infection (first part of the interval)** | | | 11.5 | 10 | 12.0 | **0.032** |
| **Time of primary infection (second part of the interval)** | | | 20.0 | 15.0 | 21.0 | 0.056 |
| **Gestational age at infection** | | | 10.0 [5.0;15.5] | 8.00 [3.5;13.0] | 12.5 [8.8;19.0] | **0.013** |
| **Type of delivery** | | | | | | 0.202 |
| | Vaginal | | 40 (75.5) | 25 (75.8) | 15 (75.0) | |
| | Cesarean-section | | 9 (17.0) | 4 (12.1) | 5 (25.0) | |
| | Instrumental | | 4 (7.6) | 4 (12.1) | 0 (0) | |
| **Gestational age at delivery** | | | 39.0 [38.0,40.0] | 39.0 [38.0;40.0] | 39.0 [38.0;40.0] | 0.362 |
| **Avidity** | | | 23.6 [18.2,31.7] | 24.0 [18.0;32.3] | 22.0 [20.0;28.0] | 0.546 |
| **Low avidity index (< 50%)** | | | 44 (80.0) | 26 (78.8) | 18 (81.8) | 1.000 |
| **Positive IgM** | | | 57 (96.6) | 34 (97.1) | 23 (95.8) | 1.000 |
| **Blood viral load** (IU/ml) Median (IQR) | | | 0.0 [0.0,0.0] | 0.0 [0.0;0.0] | 0.0 [0.0;159] | **0.003** |
| **Blood viral load** (IU/ml) Mean (SD) | | | 414 (1821) | 0.00 (0.00) | 955 (2730) | 0.231 |
| **Detectable viral load in blood** (n = 29) | | | 6/29 (20.7) | 0/16 (0) | 6/13 (46.2) | **0.004** |
| **Gestational age at blood viral load** (n = 30) | | | 22.6 [16.2,29.1] | 16.6 [14.1;26.4] | 28.4 [24.6;29.4] | **0.005** |
| **Urine viral load** (IU/ml) | | | 432 [139,3357] | 218 [0.0;424] | 2020 [484;6150] | 0.064 |
| **Detectable viral load in urine** (n = 16) | | | 12/16 (75.0) | 5/8 (62.5) | 7/8 (87.5) | 0.569 |
| **Amniocentesis** | | | 55 (91.7) | 35 (97.2) | 20 (83.3) | 0.147 |
| Positive PCR in amniotic fluid | | | 19 (34.5) | 1 (2.9) | 18 (90.0) | <**0.001** |
| Gestational age at amniocentesis | | | 21.0 [20.0,26.0] | 21.0 [20.0;26.0] | 22.5 [20.0;28.0] | 0.650 |
| **Fetal US 3rd trimester** | | | | | | 0.057 |
| Normal | | | 46 (80.7) | 31 (91.2) | 15 (65.2) | |
| CNS abnormalities | | | 2 (3.5) | 0 (0) | 2 (8.7) | |
| Non-CNS abnormalities | | | 6 (10.5) | 2 (5.9) | 4 (17.4) | |
| CNS and non-CNS abnormalities | | | 3 (5.3) | 1 (2.9) | 2 (8.7) | |
| **Fetal MRI** | | | | | | 0.389 |
| Normal | | | 11 (61.1) | 0 (0) | 11 (64.7) | |
| CNS abnormalities | | | 5 (27.8) | 1 (100) | 4 (23.5) | |

*(Continued)*

**Table 1.** (Continued)

| | All women included (n = 60) | Non-transmitter (n = 36) | Transmitter (n = 24) | p-value |
|---|---|---|---|---|
| Non-CNS abnormalities | 2 (11.1) | 0 (0) | 2 (11.8) | |
| **Hyperimmune globulin** | 20 (33.3) | 6 (16.7) | 14 (58.3) | **0.002** |
| Preventive | 6 (30.0) | 5 (83.3) | 1 (7.1) | **0.002** |
| Treatment in infected fetuses | 14 (70.0) | 1 (16.7) | 13 (92.9) | **0.002** |
| Gestational age at HIG | 23.5 [21.0,28.2] | 18.5 [13.2;22.2] | 24.8 [22.2;29.8] | **0.008** |
| **Valacyclovir** | 10 (16.7) | 1 (2.8) | 9 (37.5) | **0.001** |
| Preventive | 2 (3.3) | 1 (2.8) | 1 (4.2) | 0.200 |
| Treatment in infected fetuses | 8 (80) | 0 (0.00) | 8 (88.9) | 0.200 |
| Gestational age at valacyclovir | 24.5 [22.5,26.5] | 10.0 [10.0;10.0] | 25.0 [24.0;27.0] | 0.115 |
| **Newborn** | | | | |
| CMV congenitally infected | 20 (37.0) | 0 (0) | 20 (100) | **<0.001** |
| Abnormal physical exam | 2 (3.6) | 0 (0) | 2 (10.0) | 0.128 |
| Symptomatic at birth | 11 (57.9) | 0 (0) | 11 (57.9) | NA |

Qualitative variables are expressed as the median and IQR (interquartile range): US, Ultrasound; MRI, Magnetic Resonance Imaging; CNS, Central nervous system; PCR, Polymerase Chain Reaction; HIG, Hyperimmune globulin; NA, Not applicable; SD: Standard deviation.

were non-transmitters but 2 of the neonates had cCMV (2/36, 5.6%). Of 5 mothers on whom amniocentesis was not performed, four were transmitters (80%).

There were 20 women (33.3%) treated with CMV-HIG (200 IU/kg intravenous), 14 transmitters (14/24, 58.3%) and 6 non-transmitters (6/36, 16.7%). HIG was given after maternal infection in 6 cases (preventive treatment) and after confirmed fetal infection in 14 cases, all with positive AF PCR. Two pregnant women were treated with valacyclovir (2 g/6 hours/po) before amniocentesis (preventive treatment) and 8 women received valacyclovir if CMV-PCR in AF was positive. Finally, 20 newborns were congenitally infected by CMV (20/56, 36%), 11 of whom were symptomatic at birth (55%).

There was a broad dispersion of the CMV-CD8+IFN-γ and CMV-CD4+IFN-γ lymphocyte counts (Fig 3), and in the univariate analysis there were no significant differences among the transmitter and non-transmitter women in terms of the CMV-CD4[+]IFN-γ and CMV-CD8[+]IFN-γ counts and percentages (Table 2). Two multivariate logistic regression models were built for CD4 and CD8 responses, with other relevant clinical variables like preventive treatment with HIG/valacyclovir, and the time from maternal infection to the T-cell response included in models 1 and 2 (Table 3). Neither of these models found an association between intrauterine transmission and specific CD4 or CD8 responses (Figs 4 and 5, respectively). Nevertheless, it should be noted that in model 2 (Table 3), an interaction was found between the CD8-specific response and the time between infection to blood sampling. This model showed that the effect of the time from maternal infection to blood sample collection on IFN-γ production by CMV-CD8[+]IFN-γ lymphocytes decreased 15.1% on each passing week. Thus, the longer the time interval from infection to blood sampling, the weaker the effect of CMV-CD8[+]IFN-γ on intrauterine CMV transmission (Fig 5).

## Discussion

In this cohort of pregnant women with a primary CMV infection, we were unable to find an association between the specific CD4 and CD8 responses against CMV at the time primary infection was diagnosed and the risk of fetal transmission. Many studies have demonstrated

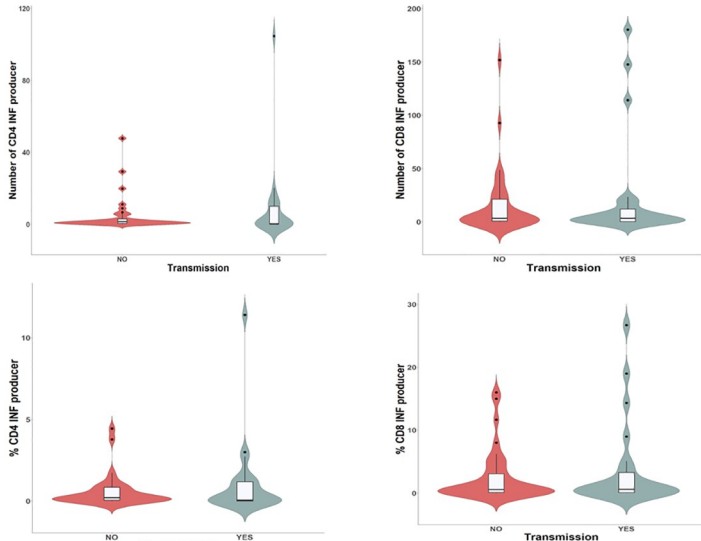

**Fig 3. T-cell mediated immune response.** Violin plot showing the lymphocyte count and the association of CMV-CD8+IFN-γ, and CMV-CD4+IFN-γ with transmission.

the essential role of T-cell immunity in controlling CMV infection. Indeed, in a cohort of pregnant women with primary CMV infection the LPR to CMV was seen to be significantly dampened or delayed in transmitter mothers [10]. Similarly, a significant delay in the development of the CD4+T-cell LPR was detected in transmitter mothers [8, 18, 19]. However, contrasting findings have also been reported, for example in a study showing that strong cellular responses to CMV in the presence of low IgG avidity was correlated with CMV transmission during primary maternal infection [11]. Also, a low IFN-γ relative response was associated with a reduction in the probability of CMV transmission in a cohort of pregnant women with a primary infection [3]. Authors hypothesized that a stronger cellular response might be correlated with longer and more intense viremia, leading to a proinflammatory environment at the placenta that facilitates virus passage [11, 20].

In some of the aforementioned studies [3, 8, 10], blood samples were collected sequentially, which may provide additional information about the temporal changes to the specific maternal T-cell response. By contrast, here we collected a single blood sample at the time of maternal infection and a distinct evolution of T-cell responses may have existed between groups that went undetected. However, if maternal specific T-cell response is to be used as a biomarker of fetal infection to help clinicians in the management of these patients, it should be an early predictive tool of fetal infection.

Also, we found that GA at maternal infection was significantly higher in transmitter than in non-transmitter mothers, in accordance with previous studies [21–23]. Risk of fetal infection increases with gestational age at maternal infection [24]. In accordance with this finding, transmitter group had a higher gestational age at maternal infection and blood viral load was evaluated later in pregnancy in this group.

Although transmission rates increased with GA, severe long-term sequelae appear to be limited to maternal infections acquired before 14 weeks' gestation [5, 25]. In primary infections during pregnancy the average rate of fetal transmission is 32% [24], similar to the rate observed in this study (40%).

CMV was detected in blood VL in 6 out of 29 women, all transmitters, finding no significant differences in urine VL between groups. When the presence of CMV-DNA in urine and

**Table 2. Results of the bivariate analysis of the T-cell immune response in women with a primary CMV infection.**

| | All women included (n = 60) | Non-transmitter (n = 36) | Transmitter (n = 24) | OR (CI 95%) | p-value |
|---|---|---|---|---|---|
| | | Median (IQR) | | | |
| Gestational age at blood sample collection | 26.0 [21.0;31.2] | 25.5 [21.0;28.2] | 26.5 [22.8;32.2] | NA | 0.290 |
| Total lymphocyte count | 2300 [1955;2620] | 2305 [2062;2665] | 2195 [1822;2502] | | 0.236 |
| % CD3-CD8$^+$ T cells | 24.6 [17.3;33.1] | 27.0 [21.1;34.1] | 23.9 [13.7;32.6] | | 0.330 |
| Total count CD3-CD8$^+$ T cells | 579 [376;775] | 579 [407;908] | 574 [291;684] | | 0.381 |
| % CMV-specific CD8$^+$IFNγ T cells | 0.58 [0.10;3.07] | 0.54 [0.11;3.07] | 0.58 [0.08;3.25] | 1.039 (0.943–1.153) | 0.988 |
| Total count CMV-specific CD8$^+$IFNγ T cells | 3.31 [0.63;18.7] | 3.31 [0.76;21.3] | 3.33 [0.24;12.1] | 1.005 (0.991–1.02) | 0.868 |
| % CD3-CD4$^+$ T cells | 36.8 [23.1;43.0] | 37.5 [28.3;42.9] | 36.8 [16.9;44.9] | NA | 0.639 |
| Total count of CD3-CD4$^+$ T cells | 778 [427;1020] | 836 [583;1030] | 701 [364;945] | | 0.196 |
| % CMV-specific CD4+IFNγ T cells | 0.15 [0.02;0.88] | 0.20 [0.05;0.84] | 0.04 [0.00;1.18] | 1.171 (0.848–1.843) | 0.251 |
| Total count CMV-specific CD4$^+$IFNγ T cells | 0.96 [0.14;5.19] | 1.44 [0.57;2.99] | 0.23 [0.00;10.0] | 1.014 (0.979–1.064) | 0.195 |
| Positive CD8 response (>0.1% IFNγ) | 43 (71.7%) | 27 (75.0%) | 16 (66.7%) | 0.667 (0.212–2.104) | 0.682 |
| Positive CD4 response (>0.1% IFNγ) | 30 (52.6%) | 19 (57.6%) | 11 (45.8%) | 0.623 (0.213–1.791) | 0.543 |

Total count and percentages of CD3-CD8$^+$ and CD3-CD4$^+$ T lymphocytes are shown, as well as the total counts and percentages of CMV-specific CD8$^+$ and CD4$^+$ T lymphocytes producing IFN-γ: IQR, Interquartile range; OR, Odds ratio; CI, Confidence Interval; NA, Not applicable (univariate analysis was not performed).

blood was studied in pregnant women with a primary infection, an association was seen between the presence of CMV-DNA in maternal blood or urine, fetal transmission [26] and congenital infection [27, 28]. Our data support the hypothesis that a detectable CMV-VL in maternal blood upon diagnosis of a primary infection could represent a relevant biomarker of intrauterine transmission. However, it should be noted that the threshold defining "detectable"/"undetectable" blood VL differs widely among participating hospitals, which may represent a limitation to the use of this parameter.

Another relevant finding was the higher proportion of children attending day-care in families of transmitter mothers. Prolonged and close contact with children <3 years of age has been associated with a higher risk of maternal infection [1]. Here, two-thirds of the cohort lived with at least one child <3 years, although the proportion of children attending day-care

**Table 3. Multivariate models of T-cell immune response.**

| Multivariate models | CD4 dichotomous analysis | | |
|---|---|---|---|
| **Model 1** | | Adjusted OR (CI 95%) | p-value |
| | CD4 Response | 0.529 (0.17–1.581) | 0.259 |
| | Preventive HIG | 0.208 (0.009–2.034) | 0.216 |
| | Time from infection (avidity) | 0.939 (0.866–1.01) | 0.1 |
| | CD4 Response by time (avidity) | 0.893 (0.734–1.056) | 0.211 |
| | **CD8 dichotomous analysis** | | |
| **Model 2** | | Adjusted OR (CI 95%) | p-value |
| | CD8 Response | 6.411 (0.475–108.898) | 0.173 |
| | Preventive HIG | 0.177 (0.008–1.715) | 0.171 |
| | Time from infection (avidity) | 1.019 (0.902–1.155) | 0.761 |
| | CD8 Response by time (avidity) | 0.849 (0.702–1.005) | 0.068 |

Results of the immune T-cells in adjusted multivariate analysis on women with a primary CMV infection for CD4 and CD8 dichotomous analysis according to the CD4 and CD8 response (positive if > 0.1%): HIG, Hyperimmune globulin; OR, Odds ratio; CI, Confidence Interval.

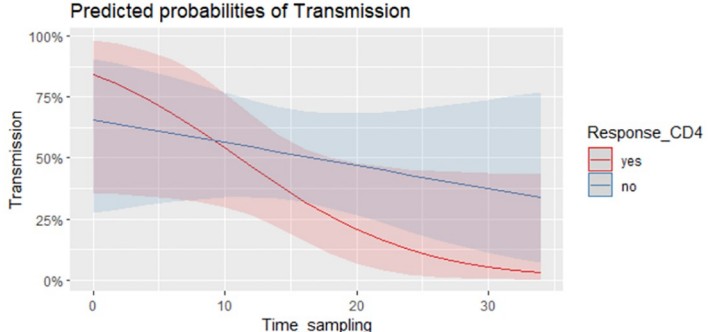

**Fig 4. CMV-CD4+IFN-γ lymphocytes relation with time between infection to blood sampling.** Model showing the time interval from infection to blood sampling against the predicted probability of CMV intrauterine transmission in women presenting a positive CD4 response (% of CD8-g-IFN producers above 0.1 in red) or not (in blue).

was significantly higher among transmitter mothers. Children attending day-care were previously seen to more often shed CMV (69%) than those cared for at home (10%) and in that study, the risk of seroconversion in parents from children shedding CMV was higher in the day-care group [29]. Thus, women with children attending day-care seem to be at a higher risk of CMV infection and of transmitting this to their fetus, although we do not have a biological explanation for this phenomenon.

In addition, it should be noted that the studies related to this work present methodological differences, since each one examines a particular area of the CMV-specific cellular response and in a different way. In recent years, the simple tools Quantiferon and ELISPOT assay have been largely used to detect antigen-specific T-cell responses. Using Quantiferon, Eldar-Yedidia et al. [3] demonstrated an association between high IFN-γ relative response to CMV and high risk of fetal transmission by measuring secretion of IFN-γ, TNF-α, IL-10 and IL-6. In addition, a higher response detected by the ELISPOT assay was shown to be associated with an increased risk of congenital infection [11]. Also, Fornara et al. investigated peptide pools of CMV proteins IE-1, IE-2, and pp65 in a cultured ELISPOT assay, the determination of which, in association with avidity index and DNAemia was found to be useful to assess the risk of fetal transmission [7]. Other studies, such as the one by Lillery and colleagues [8] was based on the CMV-specific LPR and the measurement of IFN–γ and IL–2 production by CD4+ and CD8+ T-cells. The same author in another publication investigated the membrane phenotype (CCR7

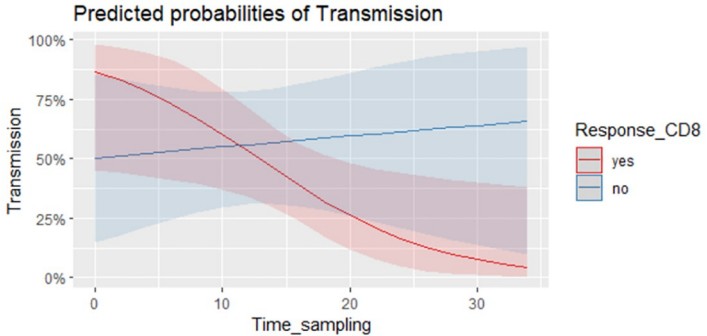

**Fig 5. Interaction time–effect of CMV-CD8+IFN-γ lymphocytes.** Model showing the time interval from infection to blood sampling against the predicted probability of CMV intrauterine transmission in women presenting a positive CD8 response (% of CD8-g-IFN producers above 0.1 in red) or not (in blue).

and CD45RA expression) of and intracellular cytokine (IFN-γ and IL-2) production by CMV-specific T-cells (stimulated with CMV-infected dendritic cells) [9]. Other authors have focused on cytotoxicity or TNF production, such as Revello et al. [10] who examined for CMV-specific CD4[+] T-cells by cytokine flow cytometry and LPR analysis frequencies of CD4[+], CD69[+], and TNF-α[+] T-cells. Thus, the wide variety of antigens, peptides and methods may justify the enormous complexity and the large magnitude that encompasses the specific T-cell response to CMV.

This study presents some limitations. First of all, sample size is limited. In order to include more patients we have performed a multicentric and prospective study in 8 hospitals. However, CMV screening during pregnancy is not mandatory in Spain and even in multicentric studies is difficult to recruit patients. Second, the time at which blood samples were collected for analysis differed among patients. Ideally, samples were obtained at the time of diagnosis of maternal infection, yet many women were referred from another center where the patient´s sample was collected. Therefore, there might have been a delay between the moment in which maternal infection was suspected and blood samples were collected. For this reason, and because the exact date of maternal infection was very challenging to establish, we have adjusted the multivariate analysis from the estimated time of maternal infection (according to the avidity test) to blood sample collection. Third, maternal blood VL was only available in a subset of women and it was therefore not included in the final multivariate model. Another limitation is that a small proportion of women were treated with HIG (n = 6) and/or valacyclovir (n = 2) prior to amniocentesis. Different treatment protocols at centers may have modified maternal viremia and they may have some influence on the risk of fetal infection, as indicated previously [30, 31]. Therefore, we decided to adjust the models based on the preventive treatment during pregnancy (HIG and/or antiviral). Finally, functional specificities of CMV-specific T-cells other than IFN-γ production (i.e cytotoxicity or TNF-alpha production) were not assessed.

In conclusion, in this cohort of pregnant women with primary CMV infection, we did not find an association between the presence of specific CD4 and CD8 responses against CMV at the time of maternal infection and the risk of fetal transmission. The detection of CMV in maternal blood at diagnosis may be considered a promising predictor of intrauterine transmission. However, further studies will be needed to better understand the role of CD4 and CD8 responses against CMV in the risk of fetal infection, and to find useful biomarkers that help us to better predict the risk of transmission during pregnancy.

## Acknowledgments

We would like to thank all the participants in this study for their kind support as well as the collaborators of the CYTRIC Study Group: Judith Hernández, Raquel Pinillos Pisón, Marie Antoinette Frick, Eneritz Velasco Arnaiz, Antoni Noguera Julian, Claudia Fortuny Guasch, María Ríos Barnés, Pablo Rojo, Cristina Epalza, Cinta Moraleda, Elisa Fernández Cooke, Luis Prieto, Jaime Carrasco, Berta Zamora, Joaquín de Vergas, Ana Martínez de Aragón, Noemí Núñez Enamorado, Rogelio Simon, Ana Camacho, Serena Villaverde, Fátima Machín, María Luz Romero, Miquel Serna, Marta Martín, Eva Dueñas, Miguel Sánchez Mateos. Also, we would like to thank Jose María Aguado for his contribution to the work and to Dr. Mark Sefton for his help in correcting the English.

## Author Contributions

**Conceptualization:** María Soriano-Ramos, Estrella Esquivel-De la Fuente, Eliseo Albert Vicent, María de la Calle, Fernando Baquero-Artigao, Sara Domínguez-Rodríguez, María Cabanes, Enery Gómez-Montes, Anna Goncé, Marta Valdés-Bango, Mª Carmen Viñuela-

Benéitez, Mar Muñoz-Chápuli Gutiérrez, Jesús Saavedra-Lozano, Irene Cuadrado Pérez, Begoña Encinas, Laura Castells Vilella, María de la Serna Martínez, Alfredo Tagarro, Paula Rodríguez-Molino, Estela Giménez Quiles, Diana García Alcázar, Antonio García Burguillo, María Dolores Folgueira, David Navarro, Daniel Blázquez-Gamero.

**Data curation:** María Soriano-Ramos, Estrella Esquivel-De la Fuente, Eliseo Albert Vicent, María de la Calle, Fernando Baquero-Artigao, Sara Domínguez-Rodríguez, María Cabanes, Enery Gómez-Montes, Anna Goncé, Marta Valdés-Bango, Mª Carmen Viñuela-Benéitez, Mar Muñoz-Chápuli Gutiérrez, Jesús Saavedra-Lozano, Irene Cuadrado Pérez, Begoña Encinas, Laura Castells Vilella, María de la Serna Martínez, Alfredo Tagarro, Paula Rodríguez-Molino, Estela Giménez Quiles, Diana García Alcázar, Antonio García Burguillo, María Dolores Folgueira, David Navarro, Daniel Blázquez-Gamero.

**Formal analysis:** María Soriano-Ramos, Estrella Esquivel-De la Fuente, Eliseo Albert Vicent, María de la Calle, Fernando Baquero-Artigao, Sara Domínguez-Rodríguez, María Cabanes, Enery Gómez-Montes, Anna Goncé, Marta Valdés-Bango, Mª Carmen Viñuela-Benéitez, Mar Muñoz-Chápuli Gutiérrez, Jesús Saavedra-Lozano, Irene Cuadrado Pérez, Begoña Encinas, Laura Castells Vilella, María de la Serna Martínez, Alfredo Tagarro, Paula Rodríguez-Molino, Estela Giménez Quiles, Diana García Alcázar, Antonio García Burguillo, María Dolores Folgueira, David Navarro, Daniel Blázquez-Gamero.

**Funding acquisition:** María Soriano-Ramos, Estrella Esquivel-De la Fuente, Sara Domínguez-Rodríguez, David Navarro, Daniel Blázquez-Gamero.

**Investigation:** María Soriano-Ramos, Estrella Esquivel-De la Fuente, Eliseo Albert Vicent, María de la Calle, Fernando Baquero-Artigao, Sara Domínguez-Rodríguez, María Cabanes, Enery Gómez-Montes, Anna Goncé, Marta Valdés-Bango, Mª Carmen Viñuela-Benéitez, Mar Muñoz-Chápuli Gutiérrez, Jesús Saavedra-Lozano, Irene Cuadrado Pérez, Begoña Encinas, Laura Castells Vilella, María de la Serna Martínez, Alfredo Tagarro, Paula Rodríguez-Molino, Estela Giménez Quiles, Diana García Alcázar, Antonio García Burguillo, María Dolores Folgueira, David Navarro, Daniel Blázquez-Gamero.

**Methodology:** María Soriano-Ramos, Estrella Esquivel-De la Fuente, Eliseo Albert Vicent, María de la Calle, Fernando Baquero-Artigao, Sara Domínguez-Rodríguez, María Cabanes, Anna Goncé, Marta Valdés-Bango, Mª Carmen Viñuela-Benéitez, Mar Muñoz-Chápuli Gutiérrez, Jesús Saavedra-Lozano, Irene Cuadrado Pérez, Begoña Encinas, Laura Castells Vilella, María de la Serna Martínez, Alfredo Tagarro, Paula Rodríguez-Molino, Estela Giménez Quiles, Diana García Alcázar, Antonio García Burguillo, María Dolores Folgueira, David Navarro, Daniel Blázquez-Gamero.

**Project administration:** María Soriano-Ramos, Estrella Esquivel-De la Fuente, Eliseo Albert Vicent, María de la Calle, Fernando Baquero-Artigao, Sara Domínguez-Rodríguez, María Cabanes, Enery Gómez-Montes, Anna Goncé, Marta Valdés-Bango, Mª Carmen Viñuela-Benéitez, Mar Muñoz-Chápuli Gutiérrez, Jesús Saavedra-Lozano, Irene Cuadrado Pérez, Begoña Encinas, Laura Castells Vilella, María de la Serna Martínez, Alfredo Tagarro, Paula Rodríguez-Molino, Estela Giménez Quiles, Antonio García Burguillo, David Navarro, Daniel Blázquez-Gamero.

**Resources:** María Soriano-Ramos, Estrella Esquivel-De la Fuente, María de la Calle, Fernando Baquero-Artigao, Sara Domínguez-Rodríguez, Enery Gómez-Montes, Anna Goncé, Mª Carmen Viñuela-Benéitez, Mar Muñoz-Chápuli Gutiérrez, Jesús Saavedra-Lozano, Irene Cuadrado Pérez, Begoña Encinas, Laura Castells Vilella, María de la Serna Martínez,

Alfredo Tagarro, Paula Rodríguez-Molino, Estela Giménez Quiles, Antonio García Burguillo, María Dolores Folgueira, David Navarro, Daniel Blázquez-Gamero.

**Software:** María Soriano-Ramos, Estrella Esquivel-De la Fuente, Sara Domínguez-Rodríguez, David Navarro, Daniel Blázquez-Gamero.

**Supervision:** María Soriano-Ramos, Estrella Esquivel-De la Fuente, Eliseo Albert Vicent, María de la Calle, Fernando Baquero-Artigao, Sara Domínguez-Rodríguez, María Cabanes, Enery Gómez-Montes, Anna Goncé, Marta Valdés-Bango, Mª Carmen Viñuela-Benéitez, Mar Muñoz-Chápuli Gutiérrez, Jesús Saavedra-Lozano, Irene Cuadrado Pérez, Begoña Encinas, Laura Castells Vilella, María de la Serna Martínez, Alfredo Tagarro, Paula Rodríguez-Molino, Estela Giménez Quiles, Diana García Alcázar, Antonio García Burguillo, María Dolores Folgueira, David Navarro, Daniel Blázquez-Gamero.

**Validation:** María Soriano-Ramos, Estrella Esquivel-De la Fuente, Eliseo Albert Vicent, María de la Calle, Fernando Baquero-Artigao, Sara Domínguez-Rodríguez, María Cabanes, Enery Gómez-Montes, Anna Goncé, Marta Valdés-Bango, Mª Carmen Viñuela-Benéitez, Mar Muñoz-Chápuli Gutiérrez, Jesús Saavedra-Lozano, Irene Cuadrado Pérez, Begoña Encinas, Laura Castells Vilella, María de la Serna Martínez, Alfredo Tagarro, Paula Rodríguez-Molino, Estela Giménez Quiles, Diana García Alcázar, Antonio García Burguillo, María Dolores Folgueira, David Navarro, Daniel Blázquez-Gamero.

**Visualization:** María Soriano-Ramos, Estrella Esquivel-De la Fuente, Eliseo Albert Vicent, María de la Calle, Fernando Baquero-Artigao, Sara Domínguez-Rodríguez, María Cabanes, Enery Gómez-Montes, Anna Goncé, Marta Valdés-Bango, Mª Carmen Viñuela-Benéitez, Mar Muñoz-Chápuli Gutiérrez, Jesús Saavedra-Lozano, Irene Cuadrado Pérez, Begoña Encinas, Laura Castells Vilella, María de la Serna Martínez, Alfredo Tagarro, Paula Rodríguez-Molino, Estela Giménez Quiles, Diana García Alcázar, Antonio García Burguillo, María Dolores Folgueira, David Navarro, Daniel Blázquez-Gamero.

**Writing – original draft:** María Soriano-Ramos, Estrella Esquivel-De la Fuente, Eliseo Albert Vicent, María de la Calle, Fernando Baquero-Artigao, Sara Domínguez-Rodríguez, María Cabanes, Enery Gómez-Montes, Anna Goncé, Marta Valdés-Bango, Mª Carmen Viñuela-Benéitez, Mar Muñoz-Chápuli Gutiérrez, Jesús Saavedra-Lozano, Irene Cuadrado Pérez, Begoña Encinas, Laura Castells Vilella, María de la Serna Martínez, Alfredo Tagarro, Paula Rodríguez-Molino, Estela Giménez Quiles, Diana García Alcázar, Antonio García Burguillo, María Dolores Folgueira, David Navarro, Daniel Blázquez-Gamero.

**Writing – review & editing:** María Soriano-Ramos, Estrella Esquivel-De la Fuente, Eliseo Albert Vicent, María de la Calle, Fernando Baquero-Artigao, Sara Domínguez-Rodríguez, María Cabanes, Enery Gómez-Montes, Anna Goncé, Marta Valdés-Bango, Mª Carmen Viñuela-Benéitez, Mar Muñoz-Chápuli Gutiérrez, Jesús Saavedra-Lozano, Irene Cuadrado Pérez, Begoña Encinas, Laura Castells Vilella, María de la Serna Martínez, Alfredo Tagarro, Paula Rodríguez-Molino, Estela Giménez Quiles, Diana García Alcázar, Antonio García Burguillo, María Dolores Folgueira, David Navarro, Daniel Blázquez-Gamero.

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
