## [Decision Letter · Decision Letter 0]

13 Jul 2022

PONE-D-22-04433The role of the T-cell mediated immune response to Cytomegalovirus infection in intrauterine transmissionPLOS ONE

Dear Dr. Soriano-Ramos,

Thank you for submitting your manuscript to PLOS ONE. After careful consideration, we feel that it has merit but does not fully meet PLOS ONE’s publication criteria as it currently stands. Therefore, we invite you to submit a revised version of the manuscript that addresses the points raised during the review process.

Please note that we have only been able to secure a single reviewer to assess your manuscript. We are issuing a decision on your manuscript at this point to prevent further delays in the evaluation of your manuscript. Please be aware that the editor who handles your revised manuscript might find it necessary to invite additional reviewers to assess this work once the revised manuscript is submitted. However, we will aim to proceed on the basis of this single review if possible. The reviewer specifically mentions that clarifications in the method and results section as well as some further statistical evaluation is required.

We look forward to receiving your revised manuscript.

Kind regards,

Thomas Tischer

Staff Editor

PLOS ONE

Journal Requirements:

“his work was supported by projects [PI 16/00807] and [PI 19/01333], from the Instituto de Salud Carlos III and co-funded by the European Regional Development Fund. DBG was supported by the Spanish Ministry of Science and Innovation -Instituto de Salud Carlos III and by Fondos FEDER “Contratos para la intensificación de actividad investigadora en el Sistema Nacional de Salud, 2020 [INT20/00086]”.

URL: https://www.isciii.es/Paginas/Inicio.aspx”

“I have read the journal's policy and the authors of this manuscript have the following competing interests: DBG received fees from MSD as speaker in educational activities. None of the remaining authors have any conflict of interests to declare.”

We note that one or more of the authors are employed by a commercial company: MSD

Reviewers' comments:

Reviewer's Responses to Questions

**Comments to the Author**

1. Is the manuscript technically sound, and do the data support the conclusions?

Reviewer #1: Partly

2. Has the statistical analysis been performed appropriately and rigorously? 

Reviewer #1: Yes

3. Have the authors made all data underlying the findings in their manuscript fully available?

Reviewer #1: No

4. Is the manuscript presented in an intelligible fashion and written in standard English?

Reviewer #1: Yes

5. Review Comments to the Author

Reviewer #1: The manuscript has important issues that need to be addressed. Methods lack important inormation and figures have low quality. Additionally, the introduction and discussion sections could improve with some more discussion. Please find the full revision in the document attached.

6. PLOS authors have the option to publish the peer review history of their article (what does this mean?). If published, this will include your full peer review and any attached files.

Reviewer #1: No

---

## [Author Response · Author response to Decision Letter 0]

10 Aug 2022

A. JOURNAL REQUESTS

• Title page with affiliations has been changed according to PLOS ONE’s style requirements. 

• Level 1, 2 and 3 Headings have been changed according to PLOS ONE's style requirements. 

• Figures: Title has been added to all figures followed by their corresponding legend. Citation of figures within the text has been changed to Fig instead of Figure, according to PLOS ONE's style requirements. 

• Tables: Title has been added to all tables. 

2. Please provide additional details regarding participant consent. In the ethics statement in the Methods and online submission information, please ensure that you have specified what type you obtained (for instance, written or verbal). If your study included minors, state whether you obtained consent from parents or guardians. If the need for consent was waived by the ethics committee, please include this information. Once you have amended this/these statement(s) in the Methods section of the manuscript, please add the same text to the “Ethics Statement” field of the submission form (via “Edit Submission”).

We have included the following statement in the Methods section “Written informed consent was requested from all women included for clinical data collecting (about their pregnancies and their newborns) and blood sampling” (lines 261-262 of the revised manuscript with track changes, page 11, 2nd paragraph). We have also included this paragraph to the ethics statement field in the Editorial Manager.

3. Please provide an amended statement that declares *all* the funding or sources of support (whether external or internal to your organization) received during this study, as detailed online in our guide for authors. Please also include the statement “There was no additional external funding received for this study.” in your updated Funding Statement. Please include your amended Funding Statement within your cover letter. We will change the online submission form on your behalf.

Grants: 

1. Grant PI 16/00807, to DBG, from Spanish Ministry of Science and Innovation Instituto de Salud Carlos III and co-funded by the European Regional Development Fund.

2. Grant 19/01333, to DBG, from Spanish Ministry of Science and Innovation Instituto de Salud Carlos III and co-funded by the European Regional Development Fund.

3. Grant INT20/00086 from Spanish Ministry of Science and Innovation Instituto de Salud Carlos III and co-funded by the European Regional Development Fund.

4. There is no additional external funding received for this study.

5. DBG received received fees from MSD as speaker in educational activities not related to the present study.

Founders (Spanish Ministry of Science and Innovation Instituto de Salud Carlos III and the European Regional Development Fund) did not play any role in study design, data collection and analysis, decision to publish or preparation of the manuscript. No commercial company funded the study or played any role in in study design, data collection and analysis, decision to publish or preparation of the manuscript. 

Sponsors websites:

www.isciii.es

https://ec.europa.eu/regional_policy/en/funding/erdf

4. a.Please provide an amended Funding Statement declaring this commercial affiliation (MSD), as well as a statement regarding the Role of Funders in your study. If the funding organization did not play a role in the study design, data collection and analysis, decision to publish, or preparation of the manuscript and only provided financial support in the form of authors' salaries and/or research materials, please review your statements relating to the author contributions, and ensure you have specifically and accurately indicated the role(s) that these authors had in your study. You can update author roles in the Author Contributions section of the online submission form.

Please also include the following statement within your amended Funding Statement. “The funder provided support in the form of salaries for authors [insert relevant initials], but did not have any additional role in the study design, data collection and analysis, decision to publish, or preparation of the manuscript. The specific roles of these authors are articulated in the ‘author contributions’ section.”

MSD is not a founder of the study.

b.Please also provide an updated Competing Interests Statement declaring this commercial affiliation along with any other relevant declarations relating to employment, consultancy, patents, products in development, or marketed products, etc. 

Within your Competing Interests Statement, please confirm that this commercial affiliation does not alter your adherence to all PLOS ONE policies on sharing data and materials by including the following statement: "This does not alter our adherence to PLOS ONE policies on sharing data and materials.” If this adherence statement is not accurate and there are restrictions on sharing of data and/or materials, please state these. 

“This does not alter our adherence to PLOS ONE policies on sharing data and materials.”

Updated funding statement

Founders (Spanish Ministry of Science and Innovation Instituto de Salud Carlos III and the European Regional Development Fund) did not play any role in study design, data collection and analysis, decision to publish or preparation of the manuscript. No commercial company funded the study or played any role in in study design, data collection and analysis, decision to publish or preparation of the manuscript. 

Grants: 

1. Grant PI 16/00807, to DBG, from Spanish Ministry of Science and Innovation Instituto de Salud Carlos III and co-funded by the European Regional Development Fund.

2. Grant 19/01333, to DBG, from Spanish Ministry of Science and Innovation Instituto de Salud Carlos III and co-funded by the European Regional Development Fund.

3. Grant INT20/00086 from Spanish Ministry of Science and Innovation Instituto de Salud Carlos III and co-funded by the European Regional Development Fund.

4. There is no additional external funding received for this study.

5. DBG received received fees from MSD as speaker in educational activities not related to the present study.

6. MSD is not a founder of the study.

Sponsors websites:

www.isciii.es

https://ec.europa.eu/regional_policy/en/funding/erdf

Updated conflict of interest statement

“This does not alter our adherence to PLOS ONE policies on sharing data and materials.”

5. In your Data Availability statement, you have not specified where the minimal data set underlying the results described in your manuscript can be found. PLOS defines a study's minimal data set as the underlying data used to reach the conclusions drawn in the manuscript and any additional data required to replicate the reported study findings in their entirety. All PLOS journals require that the minimal data set be made fully available. 

"Upon re-submitting your revised manuscript, please upload your study’s minimal underlying data set as either Supporting Information files or to a stable, public repository and include the relevant URLs, DOIs, or accession numbers within your revised cover letter. Important: If there are ethical or legal restrictions to sharing your data publicly, please explain these restrictions in detail. Note that it is not acceptable for the authors to be the sole named individuals responsible for ensuring data access.

Updated Data Availability Statement

 Data is not publicly available because is protected by European GDPR. Patient´s data is pseudoanonymized according to GDPR regulation. However, can be formally shared under a formal application and research proposal after institutional acceptance. Please send your proposal to Dr. Daniel Blázquez-Gamero (email danielblazquezgamero@gmail.com)

B. REVIEWER COMMENTS

Please see our updated Data Availability Statement

• Review Comments to the Author

Reviewer #1: 

Figures have low quality

We have improved the figure´s quality in TIFF format.

 -Introduction is too succinct, some more detail on the studies mentioned should be given. 

Introduction has been enhanced by adding more information and details about methodology and main findings of the studies mentioned through the introduction. The order of references has been changed accordingly. This is the final Introduction section included: 

“In high-income countries, around 50% of women of childbearing age are seronegative for Cytomegalovirus (CMV) [1]. However, 1-7% of these women will be infected by CMV every year, resulting in a prevalence of congenital infection of 0.14-0.7% [1, 2]. Despite the impact of CMV infection, and although it is considered the most common cause of congenital neurodevelopmental delay, several issues remain unclear. Transmission is thought to be dependent on multiple factors, such as maternal and fetal immune systems, placental factors, maternal viral, load and viral strain and the time of maternal infection [3, 4]. Timing of fetal infection is a key predictive factor for long term outcomes in children with congenital CMV, and severe sequelae are associated with fetal infection in the embryonic or early fetal period, mainly first trimester of pregnancy [5]. Risk of fetal infection during pregnancy is higher after a primary infection (32-40%) than a non-primary infection (1.4%) [4, 6] and pre-existing immune response does appear to provide some protection from fetal transmission. Nevertheless, it is still not possible to accurately predict if maternal infection will be transmitted to the fetus, and biomarkers currently available including IgG avidity index, have limited prognostic value. 

Studies on transplant recipients have also documented the importance of the CMV-specific T-cell response for the control of viral infection [7]. CMV-specific memory T-cells stimulated with peptide pools of CMV proteins IE-1, IE-2, and pp65 in a cultured enzyme-linked immunospot (ELISPOT) assay after maternal primary infection were evaluated by Fornara et al. They found that a higher cultured ELISPOT response was associated with a lower risk of transmission to the fetus [7]. 

Lillery et al investigated the specific lymphoproliferative response (LPR) and intracellular cytokine (interferon[IFN]–γ and interleukin [IL]–2) production during the first year after primary CMV infection in 49 pregnant women, finding that transmitter mothers presented a significantly delayed development of the CD4+ T-cell LPR, compared with those who did not [8]. Also, they found that the level of CMV-specific memory T-cells during the first months after infection was significantly lower in mothers who were transmitters [9].

Similarly, Revello et al analyzed specific CD4+ T-cells by cytokine flow cytometry and LPR among 74 pregnant women with primary CMV infection. This study showed that LPR to CMV was significantly lowered or delayed in transmitter mothers [10]. However, other studies have shown different results compared to the studies mentioned above. Eldar-Yedidia investigated IFN-γ secretion upon whole blood stimulation from 76 primary CMV-infected pregnant women, with either CMV-peptides or phytohemagglutinin (PHA)-mitogen. The main finding was that low IFN-γ relative response (<1.8%) strongly correlated with absence of transmission [3]. Saldan et al studied CMV ELISPOT assays in 57 pregnant women with a primary infection, finding that an increase in CMV ELISPOT levels was associated with a higher risk of fetal transmission, [11]. 

Hence, this study aimed to untangle the role of the maternal T-cell response upon diagnosis of a primary maternal infection on the risk of fetal CMV transmission.”

-Line 81 introduction: authors say that congenital infection has a prevalence of 0.14-0.7%. In their cohort 24 women out of 60 with primary infection were transmitters (40%). Is there any reason for a high prevalence of congenital CMV infection in this cohort? Could authors discuss on this matter? 

Overall congenital cytomegalovirus prevalence is 0.14-0.7% in newborns. After a primary CMV infection in pregnant women, fetal infection occurs in 32% of those pregnancies. This is a similar rate that what we found in our cohort (Leruez-Ville, AJOG 2020, reference number 24 of the revised manuscript with track changes). We have included this sentence in the Discussion section to clarify this point (lines 396 to 398 of the revised manuscript with track changes, page 22, 2nd paragraph):

“In primary infections during pregnancy the average rate of fetal transmission is 32% [24], similar to the rate observed in this study (40%).” 

-Line 93 introduction: Authors say, “the opposite effect has also been observed”. Please rephrase the sentence as studies 8 and 9 do not measure same parameters as studies 3 and 11. Also there is not enough information in some of the studies regarding the stimuli used (CMV lysate, peptides, etc.), so it is not exactly an opposite effect. 

We have deleted the sentence mentioned given the confusion. Instead, we have included the following sentence: “However, other studies have shown different results compared to the studies mentioned above.” (lines 123-124 of the revised manuscript with track changes, end of page 5, beginning of page 6). We have also explained in detail, through the introduction section, the methodology and main findings of the studies regarding maternal cellular response. 

-Line 95 introduction: I recommend changing “assess” by “untangle”, as the role has been already studied before but there are questions to clarify. 

We have changed “assess” for “untangle” in the introduction, as suggested by the reviewer (now in line 138 of the revised manuscript with track changes). 

-Line 105 methods: could authors clarify the following? “The analysis of CMV-specific CD4+ T lymphocytes was inferred from the CD8+ T-cell count and the total lymphocyte count”. 

The gating strategy is missing. A supplementary figure should be done showing the gating strategy for both CD8 and CD4 T cells, including the positive and negative control. 

We have removed that sentence and explained in detail the strategy (lines 166 to 182 of the revised manuscript with track changes, pages 7-8) in the methods section. There, we explain in greater detail the whole procedure, and we have also deleted the previous explanation. The new explanation included is the following: 

“Heparinized whole blood (0.5 ml) was simultaneously stimulated for 6 h with two sets of 15‐mer overlapping peptides (11‐mer overlap) encompassing Cytomegalovirus IE-1 and pp-65 proteins (JPT peptide Technologies GmbH (Berlin, Germany) at a concentration of 1 μg/ml per peptide, in the presence of 1 μg/ml of costimulatory monoclonal antibodies (mAbs) to CD28 and CD49d. Appropriate positive (phytohemagglutinin) and negative controls were used. Samples mock-stimulated with phosphate‐buffered saline (PBS)/dimethyl sulfoxide and costimulatory antibodies were run in parallel. Brefeldin A (10 μg/ml) was added for the last 4 h of incubation. Blood was then lysed (BD FACS lysing solution) and frozen at −80°C until tested. On the day of testing, stimulated blood was thawed at 37°C, washed, permeabilized (BD permeabilizing solution) and stained with a combination of labeled mAbs (anti‐IFNγ‐FITC, anti‐CD4‐APC-H7, anti‐CD8‐PerCP‐Cy5.5, and anti‐CD3‐APC) for 1 h at room temperature. Cells were then washed, resuspended in 200 μL of 1% paraformaldehyde in PBS, and analyzed within 2 h on an FACSCanto flow cytometer (BD Biosciences Immunocytometry Systems, San Jose, CA) using FlowJo software. CD3+/CD8+ or CD3+/CD4+ events were gated and then analyzed for CD69+/IFN‐γ production. All data were corrected for background CD69+/IFN-γ production and expressed as the percentage of cells producing CD69+/IFN-γ by the total CD8+ or CD4+ T cells (Fig 1).”

We have included an additional figure (Fig 1) showing the gating strategy for both CD8 and CD4 T cells, including the positive and negative control. The remaining figures have been designated as Fig 2, 3, 4 and 5, accordingly, to follow the appropriate order. 

In the methods, please clarify if CD28 and CD49d were added to the negative control (this should have been done to detect spontaneous unspecific stimulation). Please also explain how the response was calculated, was the background from the negative control subtracted? 

In general materials and methods section needs to be completed as information regarding some reagents used is missing (providers of CMV peptides and antibodies, clones of antibodies) 

As stated above, we have more broadly explained the methodology (see previous question and answer). 

-Figures: the quality of the figures is too low. Please ensure that data is visible. 

We have upgraded the figure in TIFF format in order to improve the quality.

-Line 273 results: “blood VL was determined earlier in pregnancy” it is not clear whether the infection took place earlier or if there was a divergence in the methodology used for sample collection. 

We have included the following sentence in discussion section (lines 390 to 393 of the revised manuscript with track changes, end of page 21, beginning of page 22): “Risk of fetal infection increases with gestational age at maternal infection [24]. In accordance with this finding, transmitter group had a higher gestational age at maternal infection and blood viral load was evaluated later in pregnancy in this group.”

Tables should have a title. Table 1, Some p-values are missing, also Blood viral load data seems to be wrong, please check as it cannot be all zeros and have a significant difference among groups. 

- We have added a title to all tables.

- Table 1, p-values (pages 13, 14 and 15): All p-values have been included and specified (also those which were referred as NS – non significant). In the case of the p-value of the variable Symptomatic at birth, it is shown as NA, since it can not be calculated because one of the groups presented 0 observations.

- Table 1, blood viral load (pages 13, 14 and 15): Blood viral load results were reported as median and interquartile range since they did not follow a normal distribution. We have indicated in the next row of the table, as requested, the values obtained with the mean and standard deviation (SD). 

Table 3, model 1 is only corrected for time from infection, no correction for treatment or response by time was included. Response by time of CD4 T cells should be also analyzed as it was done for CD8 T cells. 

Table 3 (page 20): We have included all the variables in model 1, same as in model 2. It should be noted that originally, we did not report those variables because, after applying the Akaike criteria (AIC), we only selected the variables that offered the best fit model for model 1 (CD4 T cells) and model 2 (CD8 T cels). 

Response by time of CD4 T cells should be also analyzed as it was done for CD8 T cells. 

We have included an additional figure showing the response by time of CD4 T cells (now figure 4), as it was done for CD8 T cells (now figure 5, previously fig 4). Please note that the order of the figures has been changed so as to follow the order of the models shown in table 3; that is, figure 4 refers to CD4 response by time, and figure 5 to CD8 response by time. Also, we have changed the colors of yes/no response in those figures, in order to be homogeneous (red for responders, blue for non-responders). 

-The line numbering of the discussion is missing; this complicates the reviewing process. We have included line numbering where it was missing. We apologize for the inconvenience. 

Authors conclude that there is no association between the CMV-specific T cells response and fetal transmission. The use of only two CMV peptides rather than the whole ORF should be discussed as a limitation. 

In the updated methods section, we have indicated that we did not use only two CMV peptides. Instead, we used two sets of 15–mer overlapping peptides simultaneously. You can find the explanation described in this paragraph included in the Method´s section

 (lines 166 to 170 of the revised manuscript with track changes).

“Heparinized whole blood (0.5 ml) was simultaneously stimulated for 6 h with two sets of 15‐mer overlapping peptides (11‐mer overlap) encompassing Cytomegalovirus IE-1 and pp-65 proteins (JPT peptide Technologies GmbH (Berlin, Germany) at a concentration of 1 μg/ml per peptide, in the presence of 1 μg/ml of costimulatory monoclonal antibodies (mAbs) to CD28 and CD49d”.

Also, functional analysis only considered IFNg, while cytotoxicity or TNF production were not assessed, this should be discussed as well as a limitation of the study. 

We have included this limitation at the end of the discussion section (lines 451 to 453 of the revised manuscript with track changes, page 24, 1st paragraph): “Functional specificities of CMV-specific T-cells other than IFN-γ production (i.e cytotoxicity or TNF-alpha production) were not assessed”.

 The methodological differences among all the studies related to this work should be argued further in the discussion. 

We have included an additional paragraph throughout the discussion section, explaining the methodological differences among the main studies related to this work (see lines 418 to 437 of the revised manuscript with track changes, page 23, 2nd paragraph). The new paragraph is the following:

“In addition, it should be noted that the studies related to this work present methodological differences, since each one examines a particular area of the CMV-specific cellular response and in a different way. In recent years, the simple tools Quantiferon and ELISPOT assay have been largely used to detect antigen-specific T-cell responses. Using Quantiferon, Eldar-Yedidia et al [3] demonstrated an association between high IFN-γ relative response to CMV and high risk of fetal transmission by measuring secretion of IFN-γ, TNF-α, IL-10 and IL-6; in addition, a higher response detected by the ELISPOT assay was shown to be associated with an increased risk of congenital infection [11]. Also, Fornara et al investigated peptide pools of CMV proteins IE-1, IE-2, and pp65 in a cultured ELISPOT assay, the determination of which, in association with avidity index and DNAemia was found to be useful to assess the risk of fetal transmission [7]. Other studies, such as the one by Lillery and colleagues [8] was based on the CMV-specific LPR and the measurement of IFN–γ and IL–2 production by CD4+ and CD8+ T-cells. The same author in another publication investigated the membrane phenotype (CCR7 and CD45RA expression) of and intracellular cytokine (IFN-γ and IL-2) production by CMV-specific T-cells (stimulated with CMV-infected dendritic cells) [9]. Other authors have focused on cytotoxicity or TNF production, such as Revello et al [10] who examined for CMV-specific CD4+ T-cells by cytokine flow cytometry and LPR analysis frequencies of CD4+, CD69+, and TNF-α+ T-cells. Thus, the wide variety of antigens, peptides and methods may justify the enormous complexity and the large magnitude that encompasses the specific T-cell response to CMV. ”

---

## [Decision Letter · Decision Letter 1]

23 Nov 2022

PONE-D-22-04433R1The role of the T-cell mediated immune response to Cytomegalovirus infection in intrauterine transmissionPLOS ONE

Dear Dr. Soriano-Ramos,

Thank you for submitting your manuscript to PLOS ONE. After careful consideration, we feel that it has merit but does not fully meet PLOS ONE’s publication criteria as it currently stands. Therefore, we invite you to submit a revised version of the manuscript that addresses the points raised during the review process.

We look forward to receiving your revised manuscript.

Kind regards,

Tobias Kaeser, PhD

Academic Editor

PLOS ONE

Reviewers' comments:

Reviewer's Responses to Questions

**Comments to the Author**

1. If the authors have adequately addressed your comments raised in a previous round of review and you feel that this manuscript is now acceptable for publication, you may indicate that here to bypass the “Comments to the Author” section, enter your conflict of interest statement in the “Confidential to Editor” section, and submit your "Accept" recommendation.

Reviewer #2: (No Response)

Reviewer #3: (No Response)

2. Is the manuscript technically sound, and do the data support the conclusions?

Reviewer #2: Yes

Reviewer #3: No

3. Has the statistical analysis been performed appropriately and rigorously? 

Reviewer #2: Yes

Reviewer #3: Yes

4. Have the authors made all data underlying the findings in their manuscript fully available?

Reviewer #2: No

Reviewer #3: Yes

5. Is the manuscript presented in an intelligible fashion and written in standard English?

Reviewer #2: Yes

Reviewer #3: No

6. Review Comments to the Author

Reviewer #2: Thank you for conducting and presenting this interesting study to further the understanding of CMV transmission during pregnancy. The previous review comments have been addressed satisfactorily with one exception: The data availability statement does not conform to the PLOS requirement. There must be at least one other individual/entity provided to be responsible for ensuring data access.

There are a few typographical errors in the manuscript, such as no closing parenthesis ")" in multiple places, and some missing or unnecessary punctuation (period and comma). These are very minor, but it would be helpful to readers to have them addressed.

I enjoyed reading this manuscript. Thank you for the opportunity to review.

Reviewer #3: Cytomegalovirus (CMV) is one of the most common viruses associated with congenital infection. CMV infection can be vertically transmitted to the fetus from the mother through the placenta. In this study, Ramos MS et al used a cohort of pregnant women with primary maternal CMV infection to address whether a CMV specific-T cell response in pregnancy lowers the risk of CMV intrauterine transmission. Interestingly, the authors observed no association and suggested that the presence of a detectable viral load in pregnant women with primary CMV is a possible biomarker of CMV fetal transmission. Overall, it is a very timely and important research topic that emphasizes the need for routine CMV screening in pregnant women. However, several key points remain for clarification.

1) The major limitation of the current study is the small size of the clinical cohort which limits the claims drawn from the study. The manuscript in its current state is hard to follow for non-specialists.

2) Are there any significant differences in the blood viral load between women transmitting the virus to the fetus compared to the ones that are non-transmitters?

3) If available the authors should provide information on the T cell exhaustion marker like PD-1 and whether the level of expression differs between transmitters vs non-transmitters.

7. PLOS authors have the option to publish the peer review history of their article (what does this mean?). If published, this will include your full peer review and any attached files.

Reviewer #2: **Yes: **Ruth Helmus Nissly

Reviewer #3: No

---

## [Author Response · Author response to Decision Letter 1]

28 Nov 2022

We would like to thank the reviewers for their kind suggestions and comments to the manuscript that will contribute to improve the quality of this study. Our answers to editor and reviewer´s comments can be found below. 

REVIEWERS:

Reviewer #2: Thank you for conducting and presenting this interesting study to further the understanding of CMV transmission during pregnancy. The previous review comments have been addressed satisfactorily with one exception: The data availability statement does not conform to the PLOS requirement. There must be at least one other individual/entity provided to be responsible for ensuring data access.

Data is not publicly available because is protected by European GDPR laws. Data are pseudoanonymized, and PLOS Data Availability policy includes an “exceptions to sharing materials” if “they compromise the privacy or confidentiality of human research subjects”. However, can be formally shared under a formal application and research proposal, after institutional acceptance, and we have included the PI (Dr. Daniel Blázquez-Gamero) as the contact person for the request. We have included the following statement in the methods section.

Data Availability Statement (lines 227-231 of the revised manuscript)

Data is not publicly available because is protected by European GDPR. Patient´s data is pseudoanonymized according to GDPR regulation. However, can be formally shared under a formal application and research proposal after institutional acceptance. Please send your proposal to Dr. Daniel Blázquez-Gamero (email: danielblazquezgamero@gmail.com) 

There are a few typographical errors in the manuscript, such as no closing parenthesis ")" in multiple places, and some missing or unnecessary punctuation (period and comma). These are very minor, but it would be helpful to readers to have them addressed. I enjoyed reading this manuscript. Thank you for the opportunity to review.

We have reviewed the manuscript carefully and typographical errors have been amended in the following lines of the revised manuscript:

- Closing parenthesis: lines 142 and 223.

- Deletion of period and coma: lines 223, 255 and 386.

Reviewer #3: Cytomegalovirus (CMV) is one of the most common viruses associated with congenital infection. CMV infection can be vertically transmitted to the fetus from the mother through the placenta. In this study, Ramos MS et al used a cohort of pregnant women with primary maternal CMV infection to address whether a CMV specific-T cell response in pregnancy lowers the risk of CMV intrauterine transmission. Interestingly, the authors observed no association and suggested that the presence of a detectable viral load in pregnant women with primary CMV is a possible biomarker of CMV fetal transmission. Overall, it is a very timely and important research topic that emphasizes the need for routine CMV screening in pregnant women. However, several key points remain for clarification.

1) The major limitation of the current study is the small size of the clinical cohort which limits the claims drawn from the study. The manuscript in its current state is hard to follow for non-specialists.

The initial sample size was 135 pregnant women with CMV infection. However, we selected exclusively those with a primary CMV infection (n=60) to better understand the role of time of maternal infection in T cell responses. We are aware that sample size is one of the major limitations of this study and we tried to overcome this point conducting a multicenter and prospective study in 8 tertiary hospitals. However, CMV screening during pregnancy is not mandatory in Spain to date and this is a major limitation for recruitment. We have included a sentence about sample size in Discussion section (lines 401 to 404 of the revised manuscript).

We tried to clarify definitions and explain in detail study design and laboratory methodology in the Methods section, but we are aware that this topic may be difficult for non-specialists.

2) Are there any significant differences in the blood viral load between women transmitting the virus to the fetus compared to the ones that are non-transmitters?

As indicated in table 1, the mean value of blood viral load (VL) was 955 IU/ml (SD ±2730) vs 0 IU/ml in non-transmitters (p=0.231). However, detectable VL was only present in six women, and all of them were transmitters (table 1) (p=0.004). None of the women with an undetectable VL in blood transmitted the infection to the fetus. Those results are included in table 1 and in the Results section (lines 269-270 of the revised mansucript).

3) If available the authors should provide information on the T cell exhaustion marker like PD-1 and whether the level of expression differs between transmitters vs non-transmitters.

Unfortunately, proteins such as PD-1 or other widely expressed proteins on lymphocyte populations, were not assessed in our study.

---

## [Decision Letter · Decision Letter 2]

5 Jan 2023

PONE-D-22-04433R2The role of the T-cell mediated immune response to Cytomegalovirus infection in intrauterine transmissionPLOS ONE

Dear Dr. Soriano-Ramos,

Thank you for submitting your manuscript to PLOS ONE. After careful consideration, we feel that it has merit but does not fully meet PLOS ONE’s publication criteria as it currently stands. Therefore, we invite you to submit a revised version of the manuscript that addresses the points raised during the review process.

ACADEMIC EDITOR: Please ensure you address the remaining reviewer's comment regarding the Data Availability statement. After that, the manuscript should be acceptable. Thank you!

We look forward to receiving your revised manuscript.

Kind regards,

Tobias Kaeser, PhD

Academic Editor

PLOS ONE

Journal Requirements:

Additional Editor Comments (if provided):

Please ensure you address the critique of the reviewer regarding the data availability statement. Thank you!

Reviewers' comments:

Reviewer's Responses to Questions

**Comments to the Author**

1. If the authors have adequately addressed your comments raised in a previous round of review and you feel that this manuscript is now acceptable for publication, you may indicate that here to bypass the “Comments to the Author” section, enter your conflict of interest statement in the “Confidential to Editor” section, and submit your "Accept" recommendation.

Reviewer #2: (No Response)

Reviewer #3: All comments have been addressed

2. Is the manuscript technically sound, and do the data support the conclusions?

Reviewer #2: Yes

Reviewer #3: Yes

3. Has the statistical analysis been performed appropriately and rigorously? 

Reviewer #2: Yes

Reviewer #3: Yes

4. Have the authors made all data underlying the findings in their manuscript fully available?

Reviewer #2: No

Reviewer #3: No

5. Is the manuscript presented in an intelligible fashion and written in standard English?

Reviewer #2: Yes

Reviewer #3: Yes

6. Review Comments to the Author

Reviewer #2: Thank you again for this study. I would request one change in the Data Availability statement to comply with the PLOS policy. The contact individual or organization for data requests must be a contact who is not an author of the study. Can the authors please provide the contact information for a Data Access Committee, Ethics Committee, or other body at their institution? I am sorry to belabor this point, but data accessibility is a foundational principle of PLOS journals.

from https://journals.plos.org/plosone/s/data-availability

If there are ethical or legal restrictions on sharing a sensitive data set, authors should provide the following information within their Data Availability Statement upon submission:

>Explain the restrictions in detail (e.g., data contain potentially identifying or sensitive patient information)

>Provide contact information for a data access committee, ethics committee, or other institutional body to which data requests may be sent

Please contact the journal office (plosone@plos.org) if:

>You have concerns about the ethics or legality of sharing your data

>Your institution does not have an established point of contact to field external requests for access to sensitive data

>You feel unable to share data for reasons not specified above

Why do we not allow an author to be the only point of contact for fielding requests for access to restricted data?

When possible, we recommend authors deposit restricted data to a repository that allows for controlled data access. If this is not possible, directing data requests to a non-author institutional point of contact, such as a data access or ethics committee, helps guarantee long term stability and availability of data. Providing interested researchers with a durable point of contact ensures data will be accessible even if an author changes email addresses, institutions, or becomes unavailable to answer requests.

Reviewer #3: The authors have satisfactorily addressed most of my concerns. In particular, the authors have

streamlined the manuscript for broader audience.

7. PLOS authors have the option to publish the peer review history of their article (what does this mean?). If published, this will include your full peer review and any attached files.

Reviewer #2: **Yes: **Ruth Helmus Nissly

Reviewer #3: No

---

## [Author Response · Author response to Decision Letter 2]

12 Jan 2023

JOURNAL REQUIREMENTS:

We have reviewed the reference list of the manuscript and its formatting style carefully and have not found any papers that have been retracted or mistakes regarding the style. During the whole review process, we have only added reference number 24 (during the first review in July 2022). That change was already mentioned at that moment, but any other references have been retracted. For an easier finding of references, we have included the DOI number of every paper of the reference list, in addition to traditional volume and page numbers (except for reference number 30, in which no DOI number was available). 

REVIEWERS:

Reviewer #2: Thank you again for this study. I would request one change in the Data Availability statement to comply with the PLOS policy. The contact individual or organization for data requests must be a contact who is not an author of the study. Can the authors please provide the contact information for a Data Access Committee, Ethics Committee, or other body at their institution? 

point of contact ensures data will be accessible even if an author changes email addresses, institutions, or becomes unavailable to answer requests.

Thank you for this information. As suggested by the reviewer, we have provided the contact information of the secretary of the Hospital´s Ethics Committee, María Ugalde: e-mail: mugalde.imas12@h12o.es. Data can be shared under a formal application and research proposal after institutional acceptance. We have updated the Data Availability Statement (lines 228-231 of the revised manuscript with track changes):

“Data is not publicly available because is protected by European GDPR. However, can be formally shared under a formal application and research proposal after institutional acceptance. Please send your proposal to the secretary of the 12 de Octubre Hospital Ethics Committee: María Ugalde; e-mail: mugalde.imas12@h12o.es”

Reviewer #3: The authors have satisfactorily addressed most of my concerns. In particular, the authors have

streamlined the manuscript for broader audience.

---

## [Decision Letter · Decision Letter 3]

23 Jan 2023

The role of the T-cell mediated immune response to Cytomegalovirus infection in intrauterine transmission

PONE-D-22-04433R3

Dear Dr. Soriano-Ramos,

We’re pleased to inform you that your manuscript has been judged scientifically suitable for publication and will be formally accepted for publication once it meets all outstanding technical requirements.

Kind regards,

Tobias Kaeser, PhD

Academic Editor

PLOS ONE

Additional Editor Comments (optional):

Reviewers' comments:

Reviewer's Responses to Questions

**Comments to the Author**

1. If the authors have adequately addressed your comments raised in a previous round of review and you feel that this manuscript is now acceptable for publication, you may indicate that here to bypass the “Comments to the Author” section, enter your conflict of interest statement in the “Confidential to Editor” section, and submit your "Accept" recommendation.

Reviewer #2: All comments have been addressed

2. Is the manuscript technically sound, and do the data support the conclusions?

Reviewer #2: (No Response)

3. Has the statistical analysis been performed appropriately and rigorously? 

Reviewer #2: (No Response)

4. Have the authors made all data underlying the findings in their manuscript fully available?

Reviewer #2: (No Response)

5. Is the manuscript presented in an intelligible fashion and written in standard English?

Reviewer #2: (No Response)

6. Review Comments to the Author

Reviewer #2: (No Response)

7. PLOS authors have the option to publish the peer review history of their article (what does this mean?). If published, this will include your full peer review and any attached files.

Reviewer #2: **Yes: **Ruth H. Nissly

---

## [Editor Report · Acceptance letter]

26 Jan 2023

PONE-D-22-04433R3 

The role of the T-cell mediated immune response to Cytomegalovirus infection in intrauterine transmission 

Dear Dr. Soriano-Ramos:

I'm pleased to inform you that your manuscript has been deemed suitable for publication in PLOS ONE. Congratulations! Your manuscript is now with our production department. 

Kind regards, 

on behalf of

Dr. Tobias Kaeser 

Academic Editor

PLOS ONE